# Factors That May Affect Breast Milk Macronutrient and Energy Content: A Critical Review

**DOI:** 10.3390/nu17152503

**Published:** 2025-07-30

**Authors:** Inês Rocha-Pinto, Luís Pereira-da-Silva, Diana e Silva, Manuela Cardoso

**Affiliations:** 1Faculdade de Ciências da Nutrição e Alimentação da Universidade do Porto, 4150-180 Porto, Portugal; ines.rocha.pinto7@gmail.com (I.R.-P.); dianasilva@fcna.up.pt (D.e.S.); 2Neonatology Unit, Department of Pediatrics, Hospital Dona Estefânia, Unidade Local de Saúde São José, 1169-045 Lisbon, Portugal; 3Medicine of Woman, Childhood and Adolescence Academic Area, NOVA Medical School, Universidade Nova de Lisboa, 1169-056 Lisbon, Portugal; 4CHRC—Comprehensive Health Research Centre, Nutrition Group, NOVA Medical School, Universidade Nova de Lisboa, 1169-056 Lisbon, Portugal; 5Neonatology Unit, Department of Pediatrics, Maternidade Dr. Alfredo da Costa, Unidade Local de Saúde São José, 1069-089 Lisbon, Portugal; maria.cardoso5@ulssjose.min-saude.pt; 6Nutrition Unit, Maternidade Dr. Alfredo da Costa, Unidade Local de Saúde São José, 1069-089 Lisbon, Portugal

**Keywords:** breast milk, energy content, macronutrient content, maternal factors, newborn factors, obstetrical factors, pregnancy morbidities

## Abstract

This review aimed to be comprehensive and to critically analyze the factors that may affect the macronutrient and energy content of breast milk. Systematic reviews were prioritized, even though other types of literature reviews on the subject, as well as studies not included in these reviews, were included. Reported factors that potentially affect the macronutrient and energy content of breast milk comprise: maternal factors, such as age, nutritional status, dietary intake, smoking habits, lactation stage, circadian rhythmicity, and the use of galactagogues; obstetrical factors, such as parity, preterm delivery, multiple pregnancies, labor and delivery, and pregnancy morbidities including intrauterine growth restriction, hypertensive disorders, and gestational diabetes mellitus; and newborn factors, including sexual dimorphism, and anthropometry at birth. Some factors underwent a less robust assessment, while others underwent a more in-depth analysis. For example, the milk from overweight and obese mothers has been reported to be richer in energy and fat. A progressive decrease in protein content and an increase in fat content was described over time during lactation. The milk from mothers with hypertensive disorders may have a higher protein content. Higher protein and energy content has been found in early milk from mothers who delivered prematurely.

## 1. Introduction

### 1.1. State of the Art

Advances in perinatal medicine and neonatal intensive care have improved survival rates for very preterm infants [1]. In this context, recent advances in the nutritional support of preterm infants have improved their survival rates and health outcomes both in the short [2] and long [3] term.

Nature leads to adaptations in breast milk composition in response to various factors, including maternal factors, obstetrical factors, and newborn characteristics [4,5,6]. Some of the changes in breast milk composition may occur to protect the offspring’s health [7].

Some specific factors that affect the macronutrient and energy content of breast milk have been studied more extensively.

It has been reported that maternal intake can affect the content of *n*-3 polyunsaturated fatty acids in breast milk [5,8].

Milk from overweight and obese mothers was found to be richer in energy and fat [4], although conclusive evidence for this was lacking in a recent systematic review [9]. This is important because higher maternal pre-pregnancy body mass index is associated with an increased risk of offspring overweight/obesity [10]. The composition of breast milk may contribute to the early growth trajectory of infants with obese mothers [11].

In milk from mothers who gave birth prematurely, protein content decreases and fat content increases progressively as lactation progresses, whereas energy and carbohydrate contents initially increase and then reach a steady state [12].

Milk from mothers who deliver preterm has been reported to be richer in protein and energy compared to mothers who deliver at term [7]. In particular, the milk of mothers who deliver extremely preterm is richer in protein than the milk of mothers who deliver at more advanced gestational ages [13]. This may be a nutritional advantage that supports catch-up growth in these infants [6].

Milk from mothers with hypertensive disorders and with diabetes mellitus has been described as having a higher protein content [4,13,14] and a lower fat content [4,15], respectively. The impact of changes in breast milk composition on offspring health and growth has not been explored and is an opportunity for further research [14].

In early life, female infants exhibit higher adiposity; male infants exhibit higher fat-free mass than the other sex [16]. It would be expected that breast milk for females would be richer in fat and energy to support greater fat reserves and that the milk for males would be richer in protein to support greater fat-free mass [16,17]. However, studies on sex-based differences in the composition of breast milk have produced conflicting results [18].

To discuss the assessment of the energy and macronutrient content of breast milk, it is important to address the methods used to collect and analyze the milk, as well as the usefulness of an accurate analysis for customizing the feeding of preterm infants through breast milk fortification [19].

### 1.2. Collection and Analysis of Breast Milk

The composition of breast milk is highly variable and can change depending on the stage of lactation, the time of day, and whether samples are taken before or after feeding. Therefore, the method used to collect breast milk is of utmost importance to obtain a representative sample of its composition [20]. The gold standard is to subsample and pool milk from a full breast expression at each feeding over a 24 h period. However, this approach can be impractical for mothers; therefore, alternative methods have been suggested as a more feasible option [20].

The Miris Human Milk Analyzer (Miris AB, Uppsala, Sweden) consists of a mid-infrared technology that measures the concentrations of total and true protein, fat, and carbohydrate, and calculates the total energy content based on these measurements [21]. It is currently widely used because it is more convenient; it requires less training and uses less milk than other methods [22,23]. Alternative methods are available for analyzing either all breast milk macronutrient contents or the content of each macronutrient [23,24,25].

### 1.3. Methods of Feeding Preterm Infants Based on Breast Milk

There is a consensus that the mother’s own milk is the best option for feeding very preterm infants due to its short- and long-term benefits [26,27]. For the majority of preterm infants, the nutrient content of breast milk should be adapted to the high demands of their growth [26]. For this purpose, breast milk should be supplemented with human milk fortifiers to prevent nutritional deficits and the risks of growth restriction and neurocognitive impairment [26].

Standard fortification is the most prevalent method currently used in clinical practice [19,28]. However, it assumes that the breast milk has constant protein and energy densities, carrying an inherent risk of energy-protein malnutrition [19]. Two alternative individualized fortification methods, consisting of the addition of modular macronutrient supplements to standard fortified breast milk, have been commonly used as alternatives to standard fortification [19]. In adjustable fortification, protein intake is adjusted to the infant’s metabolic response, using blood urea nitrogen as a surrogate for protein adequacy [19]. In target fortification, the content of macronutrients in breast milk is measured to determine whether to add modular macronutrient supplements to standard fortified breast milk to achieve desired nutritional targets [29,30].

### 1.4. Inaccuracy in Breast Milk Nutritional Content Estimates

Apart from target fortification, other methods estimate macronutrient and energy content of breast milk relying on data reported in the literature [12]. However, this estimate may not be accurate. It is helpful to understand the various factors that may influence breast milk composition when individualizing feeding practices for preterm infants who are exclusively fed breast milk.

### 1.5. Factors That May Affect the Macronutrient and Energy Content of Breast Milk

Systematic reviews and/or meta-analyses have evaluated the impact of maternal nutritional status and dietary intake of macronutrients on the macronutrient content of breast milk [4,5,6,7,8,9,24,25]. However, maternal and obstetrical factors, and newborn characteristics that may affect the macronutrient and energy content of breast milk, have not been extensively reviewed.

## 2. Objective

This critical review aims to gather comprehensive information on multiple factors that may affect the macronutrient and energy content of breast milk, including maternal and obstetric characteristics as well as the characteristics of the newborn.

## 3. Literature Search and Data Analysis

An extensive search of peer-reviewed articles in the databases of medical literature, including EMBASE, MEDLINE/PubMed, Cochrane Library, Scopus, Web of Science, ProQuest, Web of Science Core Collection, Scopus, Scielo, Cochrane Database of Systematic Reviews, and ClinicalTrials.gov, was performed. The search included social determinants of health as defined by the WHO [31]. Dissertations, white papers, and abstract-only papers were excluded.

The following words and word combinations were used in the search: “breast milk composition”, “breast milk”, “breast milk carbohydrate”, “breast milk energy”, “breast milk fat”, “breast milk macronutrient”, “breast milk protein”, “human milk”, “maternal factors”, “pregnancy morbidities”, “preterm delivery”, and “preterm infant”.

The literature screening and study selection were performed by two of the authors (I.R-P and L.P-d-S).

Inclusion criteria: the search was limited to the period from May 2006 to May 2025. In a pilot search, we found that, since the launch of the Miris Human Milk Analyzer in 2006 [32], this method has been the most widely used compared to its counterparts [22,23]. Therefore, to find uniformity in the methods used to determine the macronutrient and energy content of breast milk, we only included studies published after 2006. It is noteworthy that studies that used methods to determine the fatty acids, amino acids, and lactose content in breast milk were included regardless of this time criterion. The most recent systematic reviews on the subject were prioritized. Other types of literature reviews and human studies that were not included in these reviews were also included. Only articles written in English were included.

Non-inclusion criteria: any other article that did not fulfill the inclusion criteria was not included.

The critical analysis of the reviews included an evaluation of the type of review, sample size, methods used to analyze the milk macronutrient and energy content, the main results, and limitations.

The critical analysis of the original studies was partially based on the PRISMA statement [33], which was used as a reference, since this is not a systematic review. The items analyzed included: the study design, sample size, methods used to analyze the milk macronutrient and energy content, the main results, and the study limitations. Additionally, these studies were classified according to the levels of evidence (LOE): I—At least one randomized controlled trial; II-1—Well-designed cohort or case–control analytic studies; II-2—Comparisons between times or places with or without intervention; and III—Expert opinions [34].

The main text and/or tables cover the items analyzed in the reviews and original articles.

## 4. Review Results

In this review, 35 papers were included, grouped by factors (Figure 1), and then critically analyzed:-Maternal factors, including: maternal age, maternal nutritional status and dietary intake, lactation stage, circadian rhythmicity, and galactagogues use;-Obstetrical factors, including: socioeconomic status, geographical location, parity, preterm delivery, multiple pregnancy, labor, type of delivery, and pregnancy morbidities that comprise intrauterine growth restriction, hypertensive disorders, and gestational diabetes mellitus;-Neonatal factors, including sexual dimorphism and anthropometry at birth.

**Figure 1 nutrients-17-02503-f001:**
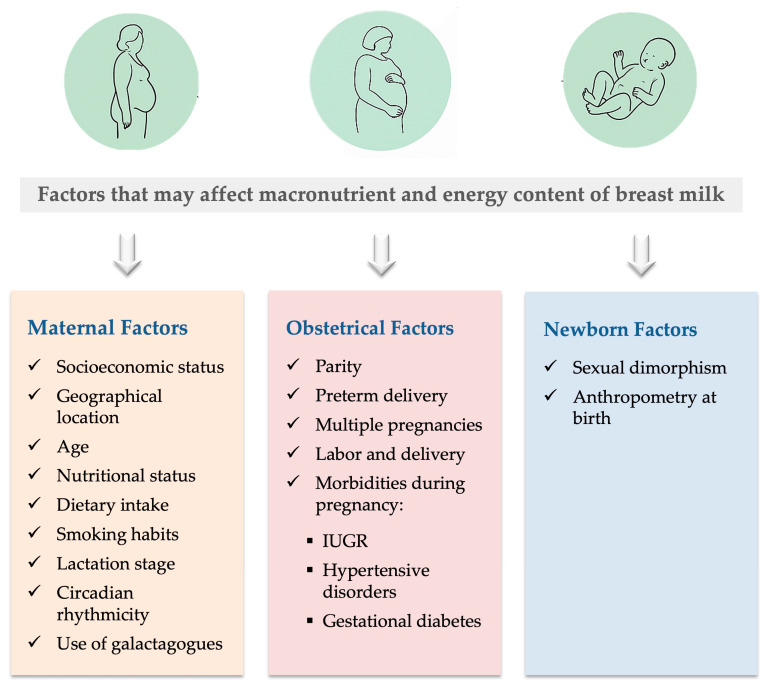
Diagram of factors that may affect macronutrient and energy content of breast milk. Abbreviation: IUGR, intrauterine growth restriction. Included images are free from copyright.

At the end of each aforementioned group, a paragraph summarizes the interpretation of the main results.

Table 1, Table 2, Table 3, Table 4, Table 5, Table 6, Table 7 and Table 8 categorize the factors that increase or decrease the macronutrient and energy content of breast milk. Energy content is addressed in Table 1 (increased) and Table 2 (decreased). Protein content is addressed in Table 3 (increased) and Table 4 (decreased). Fat content is addressed in Table 5 (increased) and Table 6 (decreased). Carbohydrate content is addressed in Table 7 (increased) and Table 8 (decreased). The studies in these tables are grouped by the type of factor and listed in chronological order of publication.

The Appendix A lists all included studies in this review and provides information on the citation, study design, sample size, analyzed factors, methods used to analyze breast milk composition, main outcomes, LOE of original studies, and limitations.

### 4.1. Maternal Factors

#### 4.1.1. Socioeconomic Status

A cross-sectional study by Chathyushya, et al. [49] investigated the total amino acid and fatty acid content of term and preterm milk, within the first week postpartum, from 120 mothers of different socioeconomic statuses. It was found that the milk from mothers with a lower socioeconomic status was significantly richer in monounsaturated and n-9 fatty acids (Table 5), while milk from mothers with an upper socioeconomic status was significantly richer in polyunsaturated, *n*-3, and *n*-6 fatty acids (Table 5).

A prospective study by Bottin, et al. [52] assessed the maternal nutritional status at delivery, diet, and breast milk composition of 48 mothers at 1, 4, 11, 18, and 25 weeks postpartum. High food insecurity indexes were significantly associated with lower fatty acid (Table 6), retinol, and amino acids contents in the breast milk. Conversely, high food insecurity indexes were significantly associated with higher lactose content in the breast milk (Table 7).

Another prospective study by Koutsiafti, et al. [43] evaluated the effect of maternal socio-demographic factors, such as age, educational level, profession, and financial status, on the breast milk composition of 50 mothers of preterm and full-term infants. The milk from mothers working in the private sector or that were self-employed had a significantly higher protein (Table 3) and fat (Table 5) content than the milk from mothers working in the public sector or that were unemployed (*p* < 0.01).

A narrative review by Samuel, et al. [50] evaluated several factors that could influence the nutritional and non-nutritional composition of breast milk. Four of the studies assessed the effect of maternal socioeconomic status (income, education, occupation, and residential area) on fat content. No consistent evidence was found except that lower socioeconomic status was associated with lower *n*-3 long-chain polyunsaturated fatty acid content (Table 6).

#### 4.1.2. Geographical Location

The aforementioned narrative review [50] included seven studies evaluating the association between the geographical location of mothers and the fatty acid content of their milk. The review found consistent evidence indicating that the fatty acid content of breast milk varies greatly by geographical location, likely due to differences in maternal diet. For example, Swedish mothers’ breast milk had a more favorable balance of fatty acids, reflected by lower linoleic acid (Table 6) and higher eicosapentaenoic acid (EPA) and docosahexaenoic acid (DHA) content compared to Chinese mothers’ breast milk (Table 5).

#### 4.1.3. Maternal Age

Prospective studies have evaluated the influence of maternal age on breast milk composition. Dritsakou, et al. [42] found that older maternal age was correlated with higher fat content in colostrum, transition, and mature milk (Table 5). Argov-Argaman, et al. [53] reported that transition milk from younger mothers (less than 37 years) had lower total fat content and higher *n*-6 fatty acids, EPA and arachidonic acid content (Table 6) than the milk from older mothers. Borràs-Novel, et al. [7] observed that breast milk from mothers older than, or equal to, 35 years was correlated with higher protein content in 1–2 weeks postpartum, compared to the milk from younger mothers (Table 3).

#### 4.1.4. Maternal Nutritional Status and Dietary Intake

Notarbartolo di Villarosa do Amaral, et al. [4], in a systematic review, included 14 studies, published between 1987 and 2016, that assessed the effects of maternal nutritional status and pregnancy morbidities (diabetes mellitus and hypertension) on breast milk nutritional composition. Of these, four studies examined the effect of maternal nutritional status on milk macronutrient content; it was found that overweight was associated with higher energy (Table 1) and fat content (Table 5).

Leghi, et al. [25], in a systematic review and meta-analysis of studies published between 1986 and 2018, evaluated the effects of maternal overweight and obesity on the content of breast milk macronutrients. A total of 31 studies were included in the qualitative synthesis; the meta-analysis of nine studies involving 872 lactating mothers suggested that maternal obesity was associated with higher fat (Table 5) and lactose content (Table 7) in breast milk, although changes depended on the stage of lactation. Most studies found a correlation of higher maternal body mass index (BMI) and/or fat mass with higher breast milk fat content. Exceptionally, the fat content in transitional milk was lower in overweight and obese mothers compared to normal-weight mothers. There was also some evidence that overweight and obese mothers had higher lactose content in colostrum. Maternal obesity measures did not appear to affect breast milk protein content. The limitations of this review include the considerable variability in the results between studies and the low quality and the lack of standardization of breast milk collection methods in many of the included studies.

A systematic review by Adhikari, et al. [24], including 50 studies published between 1954 and 2019, examined the effect of maternal dietary intake and nutritional status on breast milk macronutrient content. Higher body fat, assessed by anthropometric measures, bioelectrical impedance analysis, and dual-energy X-ray absorptiometry, was associated with higher fat content (Table 5), as reported in 68% of the studies. Lower protein content (Table 4) was found in four studies. Neither nutrient intake nor nutritional status was associated with carbohydrate content, which was assessed in 67% of the studies.

Favara, et al. [8], in a systematic review of 20 studies published between 2005 and 2023, evaluated the effects of maternal lifestyle factors on breast milk composition and infant health. Of these studies, five examined the effects of maternal dietary intake and/or nutritional status on the macronutrient content of breast milk. An association was found between higher maternal protein intake and higher breast milk content with respect to protein (Table 3), fat (Table 5), carbohydrates (Table 7) and energy (Table 1), while lower maternal fat intake was associated with higher EPA and DHA contents (Table 5). Overweight or obese mother conditions were associated with higher saturated fatty acids, *n*-6/*n*-3 ratio, and monounsaturated fatty acids content (Table 5), and with lower arachidonic acid, total *n*-3 fatty acids, alpha-linolenic acid, and DHA content (Table 6), compared to normal-weight mothers.

Another systematic review by Petersohn, et al. [5] evaluated the effect of maternal dietary intake on milk macronutrient and energy content in 27 studies published between 2015 and 2021, including 7138 mothers, and combined the results with those of the previous systematic review by Bravi et al. [56]. Higher maternal fish intake was associated with higher milk alpha-linolenic acid, DHA, and EPA contents (Table 5). Higher intake of polyunsaturated fatty acids was correlated with their higher content in milk, while higher intake of saturated fatty acids was associated with the lower content of several fatty acids (Table 6).

Hashemi Javaheri, et al. [9] evaluated the effect of maternal BMI on breast milk composition in a systematic review of 83 studies published between 1995 and 2023. Higher maternal BMI was associated with a higher breast milk *n*-6/*n*-3 fatty acids ratio (Table 5); no associations were found for other nutrients and energy contents.

Other studies with similar aims were not included in the aforementioned systematic reviews and meta-analyses [4,5,6,7,8,9,24,25].

A prospective study investigated the effect of maternal factors on breast milk energy and macronutrient content [35]. A total of 192 mothers of preterm infants were included and their milk was analyzed at 6 weeks postpartum. Maternal factors included pre-pregnancy BMI, parity, smoking habits, lactation stage, and mode of delivery. It was found that milk from mothers with a pre-pregnancy BMI at or higher than 30 kg/m^2^ was associated with higher energy and fat content (Table 1; Table 5, respectively). In normal-weight and overweight mothers, mature milk fat content decreased over time, whereas in obese mothers, it did not vary (Table 6).

In another prospective study [52], the maternal intake of meat, poultry, and fish was associated with higher total amino acid content and lower lactose content (Table 8).

A cross-sectional study evaluated the influence of maternal factors on the macronutrient and energy content of transitional breast milk [36]. A total of 159 mothers of term infants were included and their milk was analyzed at 2 weeks postpartum. The maternal factors studied included dietary intake, stress, sleep quality, and spousal support. It was found that higher breast milk energy content was correlated with a higher percentage of energy from carbohydrates and a higher percentage of energy from fat intake (Table 1). Postpartum stress, sleep quality, and spousal support were not associated with breast milk macronutrient contents.

In the aforementioned study by Borràs-Novel, et al. [7], mature milk from overweight and obese mothers was found to be associated with higher energy, protein, and fat content in mature milk over the first 4 weeks postpartum (Table 1, Table 3 and Table 5, respectively).

A cross-sectional study evaluated the influence of maternal and perinatal factors on milk nutritional composition from 181 mothers [46]. The milk from mothers with pre-gestational obesity and higher-than-recommended gestational weight gain was reported to have lower protein content compared to the milk from mothers with normal weight and appropriate gestational weight gain, respectively (Table 4).

A narrative review [14] reported that the breast milk from obese mothers had a higher total fat content, a higher ratio of *n*-6/*n*-3 fatty acids (Table 5), and higher glucose content (Table 7).

#### 4.1.5. Smoking Habits

Milk from smoking mothers was described as having significantly lower protein and fat content (Table 4; Table 6, respectively) than milk from non-smoking mothers [47]. In particular, mature milk from smoking mothers was reported to have lower fat content (Table 6) [35].

#### 4.1.6. Lactation Stage

The macronutrient content of breast milk varies at different stages of lactation, whether it is colostrum, transitional or mature milk.

In a cross-sectional study of 66 lactating mothers, protein content was found to be lower in mature milk compared to transitional milk (Table 4), while fat content was higher (Table 5) [47].

Changes in macronutrient and energy content over time have been analyzed in some prospective studies, presented below.

Burianova, et al. [35] found that protein content decreased by three weeks postpartum and then remained stable until the end of the 6th week (Table 4). Carbohydrate content increased to a stable level by the end of the 3rd week (Table 7).

Fischer Fumeaux, et al. [39] assessed the macronutrient and energy content of preterm and term breast milk from 61 mothers over time. The protein content of both preterm and term milk decreased from birth to 4 months postpartum and was lower in preterm infants at a term equivalent age (Table 4). The fat (Table 5) and energy (Table 1) content of preterm milk was higher than that of term milk during the first two weeks postpartum, whereas it was higher in term milk from three to eight weeks postpartum. Lactose content remained stable over time.

Sahin, et al. [38] examined the macronutrient and energy content of term and preterm milk from 60 mothers during the first 4 weeks of lactation. It was found that in both preterm and term milk, the protein content decreased over time (Table 4), while the fat (Table 5) and carbohydrate fat (Table 7) content increased. Energy content increased over time in term but not in preterm milk (Table 1).

Thakur, et al. [37] evaluated the association of some maternal and neonatal factors and the macronutrient content of breast milk in mothers delivering preterm infants born at or before 32 weeks of gestation, over the first 4 weeks postpartum. The lactation stage was one of the maternal factors studied that could potentially affect the macronutrient and energy content of breast milk. During that period, true protein content decreased (Table 4), fat (Table 5) and energy (Table 1) content increased, and carbohydrate content remained stable.

A meta-analysis [12] of 13 studies from a comprehensive systematic review of 23 studies suggested that, during the first 12 weeks postpartum, there is a progressive decrease in protein content (Table 4) and a progressive increase in fat content (Table 5). In contrast, energy and carbohydrate content initially increased and then reached a steady state (Table 1; Table 7, respectively).

#### 4.1.7. Circadian Rhythmicity

A systematic review by Italianer, et al. [54] examined the effect of the circadian rhythmicity on breast milk composition. A total of 83 studies involving 200 lactating mothers of term infants were included. Regarding macronutrients, a circadian variation with the acrophase in the evening was found only for total fat (Table 6). However, it should be noted that the timing of the acrophase can be biased due to differences in daylight and darkness in different countries and seasonal variations.

#### 4.1.8. Galactagogues Use

Domperidone and metoclopramide are the most commonly used and studied off-label pharmaceutical galactagogues for managing insufficient milk production [48]. Several studies have investigated the effect of galactagogues on breast milk volume; however, evidence of their effect on breast milk composition is scarce.

The study by Campbell-Yeo et al. [48] is the only that has assessed the effect of domperidone as a galactagogue on breast milk composition. In this randomized trial, milk from mothers who delivered prematurely and took domperidone had lower protein content (Table 4) and higher carbohydrate (Table 7) and calcium content compared to milk from mothers who took lactose as a placebo.

A summary of maternal factors: Among the maternal factors that may influence breast milk composition, nutritional status, dietary intake, and stage of lactation are the most studied. Systematic reviews of the effect of maternal dietary intake reported its main influence on the milk *n*-3 polyunsaturated fatty acid content. Milk from overweight and obese mothers has been reported to be richer in energy and fat compared to the milk from normal-weight mothers. However, a recent systematic review did not find conclusive evidence for these differences. Data from a comprehensive systematic review and meta-analysis of preterm milk suggest that, with the progress of lactation, a progressive decrease in protein content and a progressive increase in fat content occurs, whereas energy and carbohydrate contents initially increase and then reach a steady state.

Few studies have reported on the influence of socioeconomic status, geographical location, maternal age, smoking habits, circadian rhythmicity, and galactagogues use on the breast milk composition; therefore, further evidence is needed to confirm the reported results.

### 4.2. Obstetrical Factors

#### 4.2.1. Parity

The parity number was reported to be associated with breast milk macronutrient content. Bachour, et al. [47], reported that fat content increased in parallel with an increase in the parity number up to 3 (Table 5). Additionally, Burianova, et al. [35] found a higher protein content and lower carbohydrate content in the milk from primiparous mothers compared to multiparous mothers (Table 3 and Table 8, respectively).

Arenas, et al. [14] reported that the breast milk from multiparous mothers had a higher fat (Table 5) content than the milk from nulliparous mothers.

#### 4.2.2. Preterm Delivery

The degree of prematurity may affect the macronutrient and energy content of breast milk, as assessed in the cohort studies mentioned below.

Borràs-Novel, et al. [7] measured milk macronutrients at weeks 1, 2, 4 and 8 in 625 milk samples from 117 mothers who gave birth very prematurely (before 32 weeks of gestation). The more premature the birth, the higher the total energy and true protein content was found in the milk (Table 1; Table 3, respectively). It is worth noting that a lower gestational age was independently associated with a higher protein content in the first four weeks postpartum.

Correia, et al. [13] assessed the milk of 73 mothers of preterm infants during the first 4 weeks postpartum. Total energy and true protein content were found to be higher in the milk from mothers delivering before 28 weeks of gestation than in the milk from mothers delivering at or after that gestational age (Table 1 and Table 3, respectively).

Bauer, et al. [6] investigated the nutrient content of breast milk, from 102 mothers who gave birth prematurely, during the first eight weeks of lactation and compared it with breast milk from ten mothers who gave birth at term. Energy, fat and carbohydrate content were significantly higher in preterm than in term milk (Table 1, Table 5 and Table 7, respectively). A significantly higher protein content was found in extremely preterm milk (before 28 weeks of gestation) than in moderately preterm milk (32–33 weeks of gestation) and term milk (after 37 weeks of gestation), indicating an inverse relationship between protein content and gestational age (Table 3).

A prospective study by Dritsakou, et al. [42] measured milk energy and macronutrient contents on the 3rd, 7th and 30th day of lactation in 725 milk samples from 305 mothers who gave birth prematurely and at term. The more preterm the birth, the higher the energy, fat, and carbohydrate content in colostrum milk (Table 1, Table 5 and Table 7, respectively), the higher the energy and fat content in transitional milk (Table 1; Table 5, respectively), and the higher the content of all macronutrients and energy in mature milk (Table 1, Table 3, Table 5 and Table 7, respectively).

A systematic review and meta-analysis by Gidrewicz, et al. [40] evaluated the nutrient content of preterm and term breast milk and assessed the influence of gestational and postnatal age on the milk composition. A total of 41 studies involving 3142 lactating mothers were included. The results suggested that true protein content was higher in preterm milk than in term milk, with maximum mean differences of up to 35% in the first 3 days postpartum (Table 3); lactose content was significantly lower in preterm milk than in term milk (Table 8); fat content was not statistically different between preterm and term milk at any time during lactation.

#### 4.2.3. Multiple Pregnancy

Milk from mothers delivering multiples was found to have a lower energy and total protein content during the first 4 weeks postpartum than the milk from mothers delivering singletons (Table 2; Table 4, respectively) [7].

Another study found a weak association between single pregnancies and higher breast milk total energy content compared with multiple pregnancies [13] (Table 1).

#### 4.2.4. Labor and Delivery

It is biologically plausible that the hormonal response to labor may influence breast milk composition [7]. A prospective study found that, at four weeks postpartum, the milk from mothers who had been in labor had lower protein content than the milk from mothers who had not experienced labor [7] (Table 4).

The mode of delivery may also affect breast milk composition. A cross-sectional study [40] concluded that the milk from mothers who had a cesarean section had a higher fat content (Table 5), while milk from mothers who had a vaginal delivery had a higher carbohydrate content (Table 7). Some prospective studies have confirmed that the mode of delivery affects the macronutrient content of breast milk. Colostrum from mothers with vaginal delivery was found to have higher carbohydrate [35] (Table 7) and higher protein contents [45] (Table 3) compared with the milk from mothers with cesarean delivery. However, Sahin et al. [38] reported higher protein content in the milk from mothers who had a cesarean section (Table 3).

#### 4.2.5. Pregnancy Morbidities

##### Intrauterine Growth Restriction

Correia, et al. [13] examined the association between intrauterine growth status and breast milk macronutrient and energy content during the first 4 weeks postpartum in 127 milk samples from 73 mothers who delivered prematurely. Milk from the mothers of infants with fetal growth deceleration was found to have lower fat content than milk from mothers whose infants had normal fetal growth (Table 6). Nevertheless, this association was weak.

Borràs-Novell, et al. [7] found that milk from mothers of singleton infants with intrauterine growth restriction had a higher protein content at, or after, 4 weeks postpartum than milk from mothers of singleton infants with normal growth (Table 3).

##### Hypertensive Disorders

Hypertensive disorders that may occur during pregnancy include chronic hypertension, preeclampsia-eclampsia, preeclampsia superimposed on chronic hypertension, and gestational hypertension, i.e., transient hypertension of pregnancy or chronic hypertension identified in the latter half of pregnancy [57].

In the systematic review by Notarbartolo di Villarosa do Amaral, et al. [4], one of the studies evaluated the energy and macronutrient content of milk from mothers with hypertension. It was found that the colostrum and mature milk from these mothers had a higher total protein content compared with the milk from normotensive mothers (Table 3).

Another study [7] reported that colostrum from mothers with hypertensive disorders during pregnancy, including preeclampsia, had a lower total energy and fat content, than the milk from normotensive mothers (Table 2; Table 6, respectively).

Correia et al. [13] found that milk from mothers with chronic hypertension had higher total energy and true protein contents (Table 1; Table 3, respectively) and that mothers with gestational hypertension had higher true protein content (Table 3).

In the review by Arenas et al. [14], the milk from hypertensive mothers had higher energy, total protein, fat, and carbohydrate content compared with the milk from normotensive mothers (Table 1, Table 3, Table 5 and Table 7). In particular, the milk from mothers with gestational hypertension had lower energy and fat content than the milk from mothers without this condition (Table 2; Table 6, respectively). The milk from mothers with pre-eclampsia had a lower content of DHA (Table 6).

##### Gestational Diabetes Mellitus

Shapira, et al. [15] analyzed the macronutrient and energy content of milk from 62 mothers who delivered at term during the first 14 days postpartum. It was observed that the mature milk from mothers with gestational diabetes had lower energy (Table 2) and fat (Table 6) contents than the milk from euglycemic mothers.

A systematic review [4] concluded that milk from mothers with diabetes mellitus had lower fat (Table 6) and lactose contents (Table 8) than the milk from euglycemic mothers.

A summary of obstetrical factors: Among the obstetrical factors that may influence breast milk composition, preterm delivery and pregnancy morbidities are the most studied. Higher protein and energy content has been reported in the milk from mothers who deliver preterm compared to mothers who deliver at term. In particular, the milk of mothers who deliver extremely preterm is richer in protein than the milk from mothers who deliver at more advanced gestational ages. Among the comorbidities in pregnancy studied, a higher protein content in the milk from mothers with hypertensive disorders and a lower fat content in the milk from mothers with diabetes mellitus were observed.

Few studies have reported on the influence of parity, multiple pregnancy, and labor and delivery on the breast milk composition; therefore, further evidence is needed to confirm the reported results.

### 4.3. Neonatal Factors

#### 4.3.1. Sexual Dimorphism

Cross-sectional study studies have evaluated the influence of sexual dimorphism on breast milk macronutrient and energy contents. In a univariate analysis, Hosseini, et al. [55] found that the milk from mothers of males had significantly lower fat content than the milk from mothers of females (Table 6). Powe, et al. [41] examined potential factors affecting milk composition, including infant sex, feeding patterns, and maternal breast growth during pregnancy. After adjustment in the multiple regression analysis, milk from the mothers of males was richer in total energy than the milk from mothers of females (Table 1). Hahn, et al. [40], evaluated various maternal–infant factors, including infant sex, mode of delivery, infant birth weight and length, and postpartum days. Multivariate logistic regression analysis revealed that female sex was independently associated with higher milk carbohydrate (Table 7) and energy contents (Table 1). Quinn, et al. [58] did not find any differences in macronutrient and energy milk contents by infant sex.

In a prospective study by Fischer Fumeaux, et al. [39], several factors potentially influencing the composition of breast milk were assessed; it was found that the milk from mothers of males, whether born at term or preterm, was richer in energy and fat (Table 1; Table 5, respectively). Another prospective study by Khelouf, et al. [51] evaluated the effect of infant sex on total fat, cholesterol, carbohydrate, lactose, glucose, total protein, whey protein, casein and total energy content in the various stages of lactation, i.e., in colostrum, transitional milk, and mature milk. Colostrum and mature milk from mothers of males had lower carbohydrate and lactose contents (Table 8) and their mature milk had a higher fat content (Table 5).

Alur, et al. [17] suggest that breast milk composition for term infants may be sex-specific. The milk for males was found to have higher energy (Table 1), fat (Table 5), and carbohydrate contents (Table 7).

Finally, the narrative review by Galante, et al. [18] reported that the evidence for sex-specificity in breast milk composition is limited and conflicting.

#### 4.3.2. Anthropometry at Birth

Hahn, et al. [40], analyzed various maternal–infant factors using multivariate logistic regression analysis and found that higher birth length was independently associated with higher milk fat (Table 5) and energy (Table 1) contents.

A summary of neonatal factors: Some studies have examined how sexual dimorphism affects breast milk composition. These studies have found conflicting results regarding energy and fat content. Nevertheless, milk from the mothers of males appears to have a lower carbohydrate content. Further research is needed to confirm whether there are sex-based differences in the composition of mothers’ milk and, if confirmed, which direction they occur in.

A study concluded, through multivariate analysis, that the milk from mothers of infants born with a longer length was richer in fat and energy.

## 5. Strengths and Limitations

A strength of this review is that it compiles comprehensive and up-to-date information on all possible factors that can influence the macronutrient and energy content of breast milk. In order to be comprehensive, it included the most recent systematic reviews and other types of literature reviews on the subject, as well as studies not included in these reviews. Despite the heterogeneity of this study design, we considered it the most effective approach for building on previous knowledge. A critical analysis of the included original studies was performed, which comprised an evaluation of the study design, sample size, methods used to analyze breast milk composition, level of evidence, and limitations.

Limitations of this review should be acknowledged. First, although we subjected the original studies to critical scrutiny, the analysis and synthesis of the studies included in the reviews were predominantly based on those carried out by the respective authors. Second, the analysis of the original studies included did not follow the formal requirements of systematic reviews, including data extraction strategy, the assessment of the quality of the studies and risk of bias, and methods used for data synthesis and analysis. Adhering to these requirements would have made this review more robust.

## 6. Conclusions

This comprehensive review critically evaluated factors that may affect the macronutrient and energy contents of breast milk. These factors comprise maternal factors, obstetrical factors, as well as characteristics of the newborn.

More specifically, maternal factors include socioeconomic status, geographical location, age, nutritional status, dietary intake, smoking habits, lactation stage, circadian rhythmicity, and galactagogue use. Obstetrical factors include parity, preterm delivery, multiple pregnancies, labor and delivery, and pregnancy morbidities such as intrauterine growth restriction, hypertensive disorders, and gestational diabetes mellitus. Sexual dimorphism and anthropometry at birth are the main factors related to the newborn.

Some factors underwent a less robust assessment, while others underwent a more in-depth analysis. It is worth noting among the findings that the milk of overweight and obese mothers may be richer in energy and fat; a progressive decrease in protein content and an increase in fat content over time during lactation was described; the milk from mothers with hypertensive disorders may have a higher protein content; and higher protein and energy contents were found in the milk of mothers who delivered prematurely.

Further research is needed to confirm the significance of the reported effect of certain factors in breast milk composition, particularly those that were subjected to a less robust assessment.

## Figures and Tables

**Table 1 nutrients-17-02503-t001:** Factors that increase the total energy content of breast milk.

Factor	Study Design	Sample	Outcomes	Reference
Maternal nutritional status	Prospective, during the first 8 weeks postpartum	117 mothers delivering before 32 weeks of gestation	The mature milk from overweight and obese mothers had a higher energy content, over the first 4 weeks postpartum. LOE II-1	Borràs-Novell, et al. (2023) [7]
Maternal nutritional status	Prospective, during 6 weeks postpartum	192 mothers of preterm infants	The milk from mothers with pre-pregnancy BMI 30 kg/m^2^ or higher had a higher energy content. LOE II-1	Burianova, et al. (2019) [35]
Maternal nutritional status	Systematic review of 14 studies	Not specified	The milk from overweight mothers had a higher energy content.	Notarbartolo di Villarosa do Amaral, et al. (2019) [4]
Maternal dietary intake	Systematic review of 20 studies	Not specified	Protein intake was positively associated with milk energy content.	Favara, et al. (2024) [8]
Maternal dietary intake	Cross-sectional, at 2 weeks postpartum	159 mothers of term infants	Energy percentages from carbohydrates and fat intake were positively correlated with milk energy content. LOE II-1	Ryoo, et al. (2022) [36]
Lactation stage	Prospective, during the first 4 weeks postpartum	60 mothers delivering before or at 32 weeks of gestation	Energy content increased during the first 4 weeks of lactation. LOE II-1	Thakur, et al. (2021) [37]
Lactation stage	Prospective, during the first 4 weeks postpartum	60 mothers (39 of term and 21 of preterm infants)	Energy content increased over time in term but not in preterm milk. LOE II-1	Sahin, et al. (2020) [38]
Lactation stage	Prospective, during 4 months for preterm and 2 months for term infants	61 mothers (34 of term and 27 of preterm infants)	Energy content of preterm milk was higher than that of term milk during the first two weeks of lactation, whereas in term milk it was higher later during lactation. LOE II-1	Fischer Fumeaux, et al. (2019) [39]
Lactation stage	Systematic review and meta-analysis during the first 12 weeks postpartum	Not specified	Energy content initially increased, and then it reached a steady state in preterm milk.	Mimouni, et al. (2017) [12]
Preterm delivery	Prospective, during the first 8 weeks postpartum	117 mothers delivering before 28 weeks of gestation	The milk from mothers who delivered more prematurely had a higher total energy content than the milk from mothers who delivered at term. LOE II-1	Borràs-Novell, et al. (2023) [7]
Preterm delivery	Historical cohort, during the first 4 weeks postpartum	73 mothers delivering before 37 weeks of gestation	The degree of prematurity was positively associated with the energy content in milk, which was significantly higher in extremely preterm infants compared with very preterm infants (average: +5.95 kcal/dL; 95% CI: 2.16–9.73; *p* = 0.003). LOE II-1	Correia, et al. (2023) [13]
Preterm delivery	Prospective, during the first 8 weeks postpartum	113 mothers delivering between 23–33 weeks of gestation and 10 mothers delivering at term	The milk from mothers delivering prematurely had a significantly higher energy content than the milk from mothers who delivered at term. LOE II-1	Bauer, et al. (2011) [6]
Multiple pregnancy	Historical cohort, during the first 4 weeks postpartum	73 mothers delivering before 37 weeks of gestation	A week positive association was found between total energy content and single pregnancies (average: +3.38 kcal/dL; 95% CI: 0.07–6.83; *p* = 0.057). LOE II-1	Correia, et al. (2023) [13]
Hypertensive disorders	Narrative review	Not specified	The milk from mothers with arterial hypertension had a higher energy content than the milk from normotensive mothers.	Arenas, et al. (2025) [14]
Hypertensive disorders	Historical cohort, during the first 4 weeks postpartum	73 mothers delivering before 37 weeks of gestation	Chronic hypertension was positively associated with total energy content (average: +6.28 kcal/dL; 95% CI: 0.54–12.01; *p* = 0.034). LOE II-1	Correia, et al. (2023) [13]
Sexual dimorphism	Narrative review	Not specified	The milk from mothers of males had higher energy content than the milk from mothers of females.	Alur, et al. (2022) [17]
Sexual dimorphism	Prospective, during 4 and 2 months postpartum for preterm and term deliveries, respectively	61 mothers (34 of term and 27 of preterm infants)	Male gender was positively associated with the energy content of preterm and term milk. LOE II-1	Fischer Fumeaux, et al. (2019) [39]
Sexual dimorphism	Cross-sectional	418 mothers	The milk from mothers of females had a higher energy content (OR = 0.33, *p* = 0.017) than the milk from mothers of males. LOE II-1	Hahn, et al. (2016) [40]
Sexual dimorphism	Cross-sectional	25 mothers	The milk from mothers of males had a significantly higher energy content than the milk from mothers of females. LOE II-1	Powe, et al. (2009) [41]
Anthropometry at birth	Cross-sectional	418 mothers	The infant’s height at birth was positively associated with the milk energy content (OR = 0.74, *p* < 0.001). LOE II-1	Hahn, et al. (2016) [40]

Abbreviations: BMI, body mass index; LOE, level of evidence.

**Table 2 nutrients-17-02503-t002:** Factors that decrease the total energy content of breast milk.

Factor	Study Design	Sample	Outcomes	Reference
Preterm delivery	Prospective, during the first 8 weeks postpartum	117 mothers delivering before 28 weeks of gestation	Gestational age was negatively correlated with the milk total energy content (rho: −0.193, *p* = 0.003) content. LOE II-1	Borràs-Novell, et al. (2023) [7]
Preterm delivery	Prospective, on the 3rd, 7th and 30th day of lactation	305 mothers of preterm and term infants	Gestational age was negatively associated with the energy content in the colostrum, transitional and mature milk. LOE II-1	Dritsakou, et al. (2017) [42]
Multiple pregnancy	Prospective, during the first 8 weeks postpartum	117 mothers delivering before 32 weeks of gestation	Multiple pregnancy was negatively associated with the total energy content of milk over the first 4 weeks postpartum. LOE II-1	Borràs-Novell, et al. (2023) [7]
Hypertensive disorders	Narrative review	Not specified	Milk from mothers with gestational arterial hypertension had a lower energy content than the milk from mothers without this condition.	Arenas, et al. (2025) [14]
Hypertensive disorders	Prospective, during the first 8 weeks postpartum	117 mothers delivering before 32 weeks of gestation	Hypertensive disorders were negatively associated with the total energy content in the early milk. LOE II-1	Borràs-Novell, et al. (2023) [7]
Gestational diabetes mellitus	Prospective, at 14 days postpartum	62 mothers of term infants	Gestational diabetes mellitus was negatively associated with the milk energy content.	Shapira, et al. (2019) [15]

Abbreviation: LOE, level of evidence.

**Table 3 nutrients-17-02503-t003:** Factors that increase the total and/or true protein content of breast milk.

Factor	Study Design	Sample	Outcomes	Reference
Maternal socioeconomic status	Prospective	50 mothers of preterm and full-term infants	The milk from mothers working in the private sector or that were self-employed had a significantly higher protein content than the milk from mothers working in the public sector or that were unemployed (*p* < 0.01). LOE II-1	Koutsiafti, et al. (2021) [43]
Maternal age	Prospective, during the first 8 weeks postpartum	117 mothers delivering before 32 weeks of gestation	The milk from mothers aged 35 years or older was weakly positively correlated with the protein content at weeks 1 and 2 postpartum (r = 0.216, *p* = 0.037 and r = 0.322, *p* = 0.001, respectively), compared to younger mothers. LOE II-1	Borràs-Novell, et al. (2023) [7]
Maternal nutritional status	Prospective, during the first 8 weeks postpartum	117 mothers delivering before 32 weeks of gestation	The mature milk from overweight and obese mothers was positively associated with the protein content, over the first 4 weeks postpartum. LOE II-1	Borràs-Novell, et al. (2023) [7]
Maternal dietary intake	Systematic review of 20 studies	Not specified	Maternal protein intake was positively associated with the milk protein content.	Favara, et al. (2024) [8]
Preterm delivery	Prospective, during the first 8 weeks postpartum	117 mothers delivering before 32 weeks of gestation	The milk from mothers delivering more prematurely had a higher true protein content than the milk from mothers who delivered at term (rho: −0.307, *p* < 0.001). LOE II-1	Borràs-Novell, et al. (2023) [7]
Preterm delivery	Historical cohort, during the first 4 weeks postpartum	73 mothers delivering before 37 weeks of gestation	The degree of prematurity was positively associated with the true protein content in milk, which was significantly higher in extremely preterm infants compared with very preterm and moderate preterm infants (average +0.19 g/dL; 95% CI: 0.01–0.38; *p* = 0.043 and average +0.28 g/dL; 95% CI: 0.05–0.51; *p* = 0.017, respectively). LOE II-1	Correia, et al. (2023) [13]
Preterm delivery	Systematic review and meta-analysis of 41 studies	3142 mothers of preterm and term infants	The milk from mothers delivering prematurely had a higher true protein content than the milk from mothers delivering at term, with maximum mean differences up to 35% in the first 3 days after birth.	Gidrewicz, et al. (2014) [44]
Preterm delivery	Prospective, during the first 8 weeks postpartum	113 mothers delivering between 23–33 weeks of gestation and 10 mothers delivering at term	The milk from mothers delivering before 28 weeks of gestation had a significantly higher protein content than the milk from mothers delivering later (between 32–33 weeks of gestation or after 37 weeks of gestation). LOE II-1	Bauer, et al. (2011) [6]
Labor and delivery	Prospective, during the first 4 weeks postpartum	60 mothers (39 of term and 21 of preterm infants)	The milk from mothers who delivered by cesarean section had a higher protein content than the milk from mothers who delivered by vaginal delivery (1.794–0.848 g/dL; 1.543–0.514 g/dL respectively; *p* = 0.021). LOE II-1	Sahin, et al. (2020) [38]
Labor and delivery	Prospective, on the 2nd postpartum day	24 mothers of term infants	Vaginal delivery was associated with a higher protein content in the colostrum. LOE II-1	Dizdar, et al. (2014) [45]
Intrauterine growth restriction	Prospective, during the first 8 weeks postpartum	117 mothers delivering before 32 weeks of gestation	The milk from mothers of infants with intrauterine growth restriction had a higher protein content at or after 4 weeks postpartum than the milk from mothers of infants with normal growth. LOE II-1	Borràs-Novell, et al. (2023) [7]
Hypertensive disorders	Narrative review	Not specified	The milk from mothers with arterial hypertension had a higher protein content than the milk from normotensive mothers.	Arenas, et al. (2025) [14]
Hypertensive disorders	Historical cohort, during the first 4 weeks postpartum	73 mothers delivering before 37 weeks of gestation	Positive association with chronic hypertension and true protein content (average +0.91 g/dL; 95% CI: 0.63–1.19; *p* < 0.001) LOE II-1 Positive association with hypertension induced by pregnancy and true protein content (average +0.25 g/dL; 95% CI: 0.07–0.44; *p* = 0.007) LOE II-1	Correia, et al. (2023) [13]
Hypertensive disorders	Systematic review of 14 studies	Not specified	Hypertension disorders were positively associated with total protein content in colostrum and mature milk compared to the milk from normotensive mothers.	Notarbartolo di Villarosa do Amaral, et al. (2019) [4]
Preeclampsia	Prospective, during the first 8 weeks postpartum	117 mothers delivering before 32 weeks of gestation	Preeclampsia was positively associated with the total protein content in early milk. LOE II-1	Borràs-Novell, et al. (2023) [7]

Abbreviation: LOE, level of evidence.

**Table 4 nutrients-17-02503-t004:** Factors that decrease the total and/or true protein content of breast milk.

Factor	Study Design	Sample	Outcomes	Reference
Maternal nutritional status	Cross-sectional study	181 mothers	Pre-pregnancy obesity and gestational weight gain were associated with a lower protein content in milk compared to eutrophic mothers and those with adequate gestational weight gain, respectively. LOE II-1	Marano, et al. (2023) [46]
Maternal nutritional status	Systematic review of 50 studies	Not specified	Higher body fat was associated with a lower milk protein content.	Adhikari, et al. (2022) [24]
Smoking habits	Cross-sectional	66 mothers	Smoking habits were negatively associated with milk protein content. LOE II-1	Bachour, et al. (2012) [47]
Lactation stage	Prospective, during the first 4 weeks postpartum	60 mothers delivering before or at 32 weeks of gestation	True protein content decreased during the first 4 weeks of lactation. LOE II-1	Thakur, et al. (2021) [37]
Lactation stage	Prospective, during the first 4 weeks postpartum	60 mothers (39 of term and 21 of preterm infants)	The milk protein content decreased over time in the milk from mothers delivering prematurely and at term. LOE II-1	Sahin, et al. (2020) [38]
Lactation stage	Prospective, during 4 and 2 months postpartum for preterm and term deliveries, respectively	61 mothers (34 of term and 27 of preterm infants)	The protein content of both preterm and term milk decreased from birth to four months postpartum. Although there were no significant differences in the protein content between the two groups during this period, preterm milk had a lower protein content when reached term equivalent age. LOE II-1	Fischer Fumeaux, et al. (2019) [39]
Lactation stage	Prospective, during 6 weeks postpartum	192 mothers of preterm infants	The milk protein content decreased for the first three weeks postpartum and then remained stable until the end of the 6th week. LOE II-1	Burianova, et al. (2019) [35]
Lactation stage	Systematic review and meta-analysis of preterm milk composition during the first 12 weeks postpartum	Not specified	There was a progressive decrease in milk total protein content.	Mimouni, et al. (2017) [12]
Lactation stage	Cross-sectional	66 mothers	Protein content was lower in mature milk than in transitional milk. LOE II-1	Bachour, et al. (2012) [47]
Galactagogues	Clinical trial, at 7 and 14 days postpartum	46 mothers	The milk from mothers who used domperidone had a lower protein content than the milk from mothers who used placebo. LOE I	Campbell-Yeo et al. (2010) [48]
Parity	Prospective, during 6 weeks postpartum	192 mothers of preterm infants	Parity was negatively associated with the milk protein content. LOE II-1	Burianova, et al. (2019) [35]
Preterm delivery	Prospective, on the 3rd, 7th and 30th day of lactation	305 mothers of preterm and term infants	The more preterm the birth, the higher the protein content in mature milk. LOE II-1	Dritsakou, et al. (2017) [42]
Multiple pregnancy	Prospective, during the first 8 weeks postpartum	117 mothers delivering before 32 weeks of gestation	Multiple pregnancy was negatively associated with the total protein content in milk over the first 4 weeks postpartum. LOE II-1	Borràs-Novell, et al. (2023) [7]
Labor and delivery	Prospective, during the first 8 weeks postpartum	117 mothers delivering before 32 weeks of gestation	Labor before birth was associated with a lower protein content of milk at 4 weeks postpartum. LOE II-1	Borràs-Novell, et al. (2023) [7]

Abbreviation: LOE, level of evidence.

**Table 5 nutrients-17-02503-t005:** Factors that increase the fat content of breast milk.

Factor	Study Design	Sample	Outcomes	Reference
Maternal socioeconomic status	Cross-sectional	120 mothers of term and preterm infants	The milk from mothers with a lower socioeconomic status was significantly richer in monounsaturated and n-9 fatty acids, while mothers with an upper socioeco-nomic status was significantly richer in polyunsaturated, *n*-3, and *n*-6 fatty acids. LOE II-1	Chathyushya, et al. (2023) [49]
Maternal socioeconomic status	Prospective	50 mothers of preterm and full-term infants	The milk from mothers working in the private sector or that were self-employed had a significantly higher fat content than the milk from mothers working in the public sector or that were unemployed (*p* < 0.01). LOE II-1	Koutsiafti, et al. (2021) [43]
Geographical location	Narrative review	Not specified	Swedish mothers’ breast milk had higher EPA and DHA content compared to Chinese mothers’ breast milk.	Samuel, et al. (2020) [50]
Maternal age	Prospective, on the 3rd, 7th and 30th day of lactation	305 mothers of pre-term and term infants	Maternal age was positively correlated with the fat content in colostrum, transitional and mature milk. LOE II-1	Dritsakou, et al. (2017) [42]
Maternal nutritional status	Systematic review of 83 studies	11,310 mothers	Higher maternal BMI was associated with a higher ratio of *n*-6/*n*-3 polyunsaturated fatty acids.	Hashemi Javaheri, et al. (2025) [9]
Maternal nutritional status	Narrative review	Not specified	The milk from obese mothers had a higher total fat content and a higher ratio of *n*-6/*n*-3 polyunsaturated fatty acids.	Arenas, et al. (2025) [14]
Maternal nutritional status	Systematic review of 20 studies	Not specified	The milk from overweight and obese mothers had a higher saturated fatty acids content, a higher *n*-6/*n*-3 ratio, and a higher monounsaturated fatty acids content, compared to normal-weight mothers.	Favara, et al. (2024) [8]
Maternal nutritional status	Prospective, during the first 8 weeks postpartum	117 mothers delivering before 32 weeks of gestation	Mature milk from overweight and obese mothers had a higher fat content over the first 4 weeks postpartum. LOE II-1	Borràs-Novell, et al. (2023) [7]
Maternal nutritional status	Systematic review of 50 studies	Not specified	Higher maternal body fat was associated with a higher milk fat content	Adhikari, et al. (2022) [24]
Maternal nutritional status	Systematic review and meta-analysis	5078 lactating mothers in the review, 872 in the meta-analysis	Maternal overweight and obesity were positively correlated with milk fat content. However, in transitional milk, the fat content was lower in overweight/obese mothers than in normal-weight mothers.	Leghi, et al. (2020) [25]
Maternal nutritional status	Prospective, during 6 weeks postpartum	192 mothers of preterm infants	The milk from mothers with a pre-pregnancy BMI of 30 kg/m^2^ or higher had a higher fat content. LOE II-1	Burianova, et al. (2019) [35]
Maternal nutritional status	Systematic review of 14 studies	Not specified	The milk from overweight mothers had a higher fat content.	Notarbartolo di Villarosa do Amaral, et al. (2019) [4]
Maternal dietary intake	Systematic review of 27 studies	7138 mothers	Maternal fish intake was positively associated with the ALA, DHA, and EPA content in milk.	Petersohn, et al. (2024) [5]
Maternal dietary intake	Systematic review of 27 studies	7138 mothers	Maternal PUFAs intake was positively correlated with the PUFAs content in milk.	Petersohn, et al. (2024) [5]
Maternal dietary intake	Systematic review of 20 studies	Not specified	Maternal protein intake was positively associated with the milk fat content.	Favara, et al. (2024) [8]
Lactation stage	Prospective, during the first 4 weeks postpartum	60 mothers delivering before or at 32 weeks of gestation	The milk fat content increased during the first 4 weeks of lactation. LOE II-1	Thakur, et al. (2021) [37]
Lactation stage	Prospective, during the first 4 weeks postpartum	60 mothers (39 of term and 21 of preterm infants)	Fat content increased both in preterm and term milk over lactation time. LOE II-1	Sahin, et al. (2020) [38]
Lactation stage	Prospective, during 4 and 2 months postpartum for preterm and term deliveries, respectively	61 mothers (34 of term and 27 of preterm infants)	Fat content of preterm milk was higher than of term milk in the first two weeks of lactation, whereas fat content in term milk was higher later in the lactation period. LOE II-1	Fischer Fumeaux, et al. (2019) [39]
Lactation stage	Systematic review and meta-analysis of preterm milk composition during the first 12 weeks postpartum	Not specified	The milk fat content progressively increased during lactation.	Mimouni, et al. (2017) [12]
Lactation stage	Cross-sectional	66 mothers	Fat content was higher in mature milk than in transitional milk. LOE II-1	Bachour, et al. (2012) [47]
Parity	Narrative review	Not specified	The milk from multiparous mothers had a higher fat content than the milk from nulliparous mothers.	Arenas, et al. (2025) [14]
Parity	Cross-sectional	66 mothers	The milk fat content increased in parallel with an increase in parity number up to 3. LOE II-1	Bachour, et al. (2012) [47]
Preterm delivery	Prospective, during the first 8 weeks postpartum	113 mothers delivering between 23–33 weeks of gestation and 10 mothers delivering at term	The milk from mothers delivering before 28 weeks of gestation had a significantly higher fat content than the milk from mothers delivering at term. LOE II-1	Bauer, et al. (2011) [6]
Labor and delivery	Cross-sectional	418 mothers	The milk from mothers who had a cesarean section had a higher fat content than the milk from mothers who delivered vaginally (OR = 2.47, *p* < 0.001). LOE II-1	Hahn, et al. (2016) [40]
Hypertensive disorders	Narrative review	Not specified	The milk from mothers with arterial hypertension had a higher fat content compared with normotensive mothers.	Arenas, et al. (2025) [14]
Sexual dimorphism	Prospective, during 720 days postpartum	92 mothers	Colostrum and mature milk from mothers of males had a higher fat content than the milk from mothers of females. LOE II-1	Khelouf, et al. (2023) [51]
Sexual dimorphism	Narrative review	Not specified	The milk from mothers of males had higher fat content than the milk from mothers of females.	Alur, et al. (2022) [17]
Sexual dimorphism	Prospective, during 4 and 2 months postpartum for preterm and term deliveries, respectively	61 mothers (34 of term and 27 of preterm infants)	The milk from mothers of males had a higher fat content, both in preterm and term milk. LOE II-1	Fischer Fumeaux, et al. (2019) [39]
Anthropometry at birth	Cross-sectional	418 mothers	Infant’s height at birth was positively associated with the milk fat content (OR = 0.84, *p* = 0.004). LOE II-1	Hahn, et al. (2016) [40]

Abbreviations: ALA, alpha-linolenic acid; BMI, body mass index; DHA, docosahexaenoic acid; EPA, eicosapentaenoic acid; LOE, level of evidence.

**Table 6 nutrients-17-02503-t006:** Factors that decrease the fat content of breast milk.

Factor	Study Design	Sample	Outcomes	Reference
Maternal socioeconomic status	Prospective, at weeks 1, 4, 11, 18, and 25 postpartum	48 mothers	High food insecurity indexes were significantly associated with lower fatty acid (aß-coef = −7.2, *p* value = 0.03). LOE II-1	Bottin, et al. (2022) [52]
Maternal socioeconomic status	Narrative review	Not specified	No consistent evidence of the effect of socioeconomic status on breast milk composition was found except that lower socioeconomic status was associated with lower *n*-3 long-chain polyunsaturated fatty acid content.	Samuel, et al. (2020) [50]
Geographical location	Narrative review	Not specified	Swedish mothers’ breast milk had a lower linoleic acid content compared to Chinese mothers’ breast milk.	Samuel, et al. (2020) [50]
Maternal age	Prospective, 14 days postpartum	49 mothers	The transition milk from younger mothers had a lower total fat content and a higher *n*-6 fatty acids, EPA and ARA content than that from older mothers. LOE II-1	Argov-Argaman, et al. (2016) [53]
Maternal nutritional status	Systematic review of 20 studies	Not specified	Overweight or obese mother conditions were negatively associated with milk ARA, total *n*-3 fatty acids, ALA, and DHA content, compared to normal-weight mothers.	Favara, et al. (2024) [8]
Maternal nutritional status	Prospective, during 6 weeks postpartum	192 mothers of preterm infants	Fat content in mature milk decreased overtime, both in normal-weight and overweight mothers. LOE II-1	Burianova, et al. (2019) [35]
Maternal dietary intake	Systematic review of 27 studies	7138 mothers	The saturated fatty acids intake was negatively associated with several fatty acids content in milk.	Petersohn, et al. (2024) [5]
Maternal dietary intake	Systematic review of 20 studies	Not specified	The fat intake was negatively associated with the milk EPA and DHA content.	Favara, et al. (2024) [8]
Smoking habits	Prospective, during 6 weeks postpartum	192 mothers of preterm infants	Smoking habits were negatively associated with the fat content in mature milk (*p* = 0.026). LOE II-1	Burianova, et al. (2019) [35]
Smoking habits	Cross-sectional	66 mothers	Smoking habits were negatively associated with the fat content in milk.	Bachour, et al. (2012) [47]
Circadian rhythmicity	Systematic review of 83 studies	200 mothers of term infants	There is a circadian variation in milk total fat content with the acrophase in the evening. LOE II-1	Italianer, et al. (2020) [54]
Preterm delivery	Prospective, on the 3rd, 7th and 30th day of lactation	305 mothers of preterm and term infants	The more preterm the birth, the higher the fat content in colostrum, transitional, and mature milk. LOE II-1	Dritsakou, et al. (2017) [42]
Intrauterine growth restriction	Historical cohort, during the first 4 weeks postpartum	73 mothers delivering before 37 weeks of gestation	Fat content was weakly and negatively associated with IUGR, both in SGA infants and AGA infants with fetal growth deceleration (average 0.44 g/dL; 95% CI: 0.92 to 0.05; *p* = 0.079 and average 0.36 g/dL; 95% CI: 0.74 to 0.02; *p* = 0.066, respectively). LOE II-1	Correia, et al. (2023) [13]
Hypertensive disorders	Narrative review	Not specified	The milk from mothers with gestational hypertension had a lower fat content compared with milk from normotensive mothers.	Arenas, et al. (2025) [14]
Hypertensive disorders	Narrative review	Not specified	The milk from mothers with preeclampsia was associated with a lower DHA content.	Arenas, et al. (2025) [14]
Hypertensive disorders	Prospective, during the first 8 weeks postpartum	117 mothers delivering before 32 weeks of gestation	The early milk from mothers with hypertensive disorders during pregnancy, including preeclampsia, had a lower fat content than that from normotensive mothers. LOE II-1	Borràs-Novell, et al. (2023) [7]
Diabetes mellitus	Systematic review of 14 studies	Not specified	Diabetes mellitus was negatively associated with the milk fat content.	Notarbartolo di Villarosa do Amaral, et al. (2019) [4]
Gestational diabetes mellitus	Prospective, at 14 days postpartum	62 mothers of term infants	Gestational diabetes mellitus was negatively associated with the milk fat content. LOE II-1	Shapira, et al. (2019) [15]
Sexual dimorphism	Cross-sectional	119 mothers	The milk from mothers of males had a significantly lower fat content than the milk from mothers of females. LOE II-1	Hosseini, et al. (2020) [55]

Abbreviations: AGA, appropriate for gestational age; ALA, alpha-linolenic acid; ARA, arachidonic acid; DHA, docosahexaenoic acid; EPA, eicosapentaenoic acid; IUGR, intrauterine growth restriction; LOE, level of evidence; SGA, small for gestational age.

**Table 7 nutrients-17-02503-t007:** Factors that increase the carbohydrate content in breast milk.

Factor	Study Design	Sample	Outcomes	Reference
Maternal socioeconomic status	Prospective, at weeks 1, 4, 11, 18, and 25 postpartum	48 mothers	High food insecurity indexes were significantly associated with higher lactose content in the breast milk. LOE II-1	Bottin, et al. (2022) [52]
Maternal nutritional status	Narrative review	Not specified	The milk from obese mothers was associated with a higher glucose content.	Arenas, et al. (2025) [14]
Maternal nutritional status	Systematic review and meta-analysis	5078 lactating mothers included in the review, and 872 in the meta-analysis	Maternal BMI and/or fat mass was positively correlated with the milk lactose content. Of note, lactose content may depend on the stage of lactation, with a higher content found in the colostrum.	Leghi, et al. (2020) [25]
Maternal dietary intake	Systematic review of 20 studies	Not specified	Maternal protein intake was positively associated with the milk carbohydrate content.	Favara, et al. (2024) [8]
Lactation stage	Prospective, during the first 4 weeks postpartum	60 mothers (39 of term and 21 of preterm infants)	Carbohydrate content of preterm and term milk increased over time. LOE II-1	Sahin, et al. (2020) [38]
Lactation stage	Prospective, during 6 weeks postpartum	192 mothers of preterm infants	Lactation stage was positively associated with the milk carbohydrate content until the end of 3rd week postpartum. LOE II-1	Burianova, et al. (2019) [35]
Lactation stage	Systematic review and meta-analysis of preterm milk composition during the first 12 weeks postpartum	Not specified	The milk carbohydrate content initially increased and then reached a steady state.	Mimouni, et al. (2017) [12]
Galactagogues	Clinical trial, at 7 and 14 days postpartum	46 mothers	The milk from mothers who used domperidone had a higher carbohydrate content than in milk from those who used placebo. LOE I	Campbell-Yeo et al. (2010) [48]
Parity	Prospective, during 6 weeks postpartum	192 mothers of preterm infants	Parity was positively correlated with the carbohydrate content in colostrum. LOE II-1	Burianova, et al. (2019) [35]
Preterm delivery	Prospective, during the first 8 weeks postpartum	113 mothers delivering between 23–33 weeks of gestation and 10 mothers delivering at term	The milk carbohydrate content was significantly higher in the milk from mothers delivering before 28 weeks of gestation than milk from mothers delivering at term. LOE II-1	Bauer, et al. (2011) [6]
Labor and delivery	Prospective, during 6 weeks postpartum	192 mothers of preterm infants	Vaginal delivery was positively associated with the colostrum carbohydrate content (*p* = 0.021). LOE II-1	Burianova, et al. (2019) [35]
Labor and delivery	Cross-sectional	418 mothers	Vaginal delivery was positively associated with the milk carbohydrate content (OR = 0.50, *p* = 0.005). LOE II-1	Hahn, et al. (2016) [40]
Hypertensive disorders	Narrative review	Not specified	The milk from mothers with hypertension had a higher carbohydrate content, compared with the milk from normotensive mothers.	Arenas, et al. (2025) [14]
Sexual dimorphism	Narrative review	Not specified	The milk from mothers of males had higher carbohydrate content the milk from mothers of females.	Alur, et al. (2022) [17]
Sexual dimorphism	Cross-sectional	418 mothers	The milk from mothers of females had a higher carbohydrate content (OR = 0.56, *p* = 0.012). LOE II-1	Hahn, et al. (2016) [40]

Abbreviation: LOE, level of evidence.

**Table 8 nutrients-17-02503-t008:** Factors that decrease the carbohydrate content in breast milk.

Factor	Study Design	Sample	Outcomes	Reference
Maternal dietary intake	Prospective, at weeks 1, 4, 11, 18, and 25 postpartum	48 mothers	Intake of meat, poultry, and fish was associated with lower lactose content (aß-coef = −15.6, *p* value = 0.01).	Bottin, et al. (2022) [52]
Preterm delivery	Prospective, on the 3rd, 7th and 30th day of lactation	305 mothers of pre-term and term infants	Gestational age was negatively associated with carbohydrate content in colostrum and mature milk. LOE II-1	Dritsakou, et al. (2017) [42]
Preterm delivery	Systematic review and meta-analysis of 41 studies	3142 mothers of preterm and term infants	The lactose content was significantly lower in the milk from mothers delivering prematurely than in the milk from mothers delivering at term.	Gidrewicz, et al. (2014) [44]
Diabetes mellitus	Systematic review of 14 studies	Not specified	Diabetes mellitus was negatively associated with the milk lactose content.	Notarbartolo di Villarosa do Amaral, et al. (2019) [4]
Sexual dimorphism	Prospective, during 720 days postpartum	92 mothers	Colostrum and mature milk from mothers of males had a lower carbohydrate and lactose content. LOE II-1	Khelouf, et al. (2023) [51]

Abbreviation: LOE, level of evidence.

## Data Availability

Not applicable.

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
