# Peer review of "Factors That May Affect Breast Milk Macronutrient and Energy Content: A Critical Review"

_nutrients, 2025, doi:10.3390/nu17152503_

Round 1
Reviewer 1 Report
Comments and Suggestions for Authors
This article is a comprehensive overview of articles that mainly deal with the influence of various factors on the composition of breast milk. The authors analyse in detail how maternal factors such as age, nutritional status, feeding habits, smoking and stage of lactation as well as obstetric factors such as prematurity, multiple pregnancies and pregnancy complications affect the composition of milk. The review also takes into account the influence of neonatal factors such as gender and anthropometry at birth on the nutrient profile of breastmilk.
The paper is a comprehensive summary of previous studies.
The diversity of the databases used is noteworthy.
For the correct results, it was also important to use the amount of research from less than 20 years to standardise the research methodology. The results obtained are presented in the form of several tables with the most important information.
My objectives/suggestions:
1. Can the authors please present a graphical "literature search scheme" (flowchart) in section "3. Literature search"?
2. Optional: a simple summarising graphic would be a good addition to the paper.
Author Response
1. Can the authors please present a graphical "literature search scheme" (flowchart) in section "3. Literature search"?
Response: Thank you for your question. According to the reviewer's suggestion, we included the PRISMA statement as a reference for our review, even though it is not a systematic review. The original study design did not include all the mandatory steps of the PRISMA statement. Consequently, the number of titles and abstracts screened during the identification stage was not recorded, so it is now impossible to create an accurate study selection flowchart. Nevertheless, in the revised section 3. “Literature Search and Data Analysis” we have specified that the literature screening and study selection were performed by two of the authors (I.R-P and L.P-d-S) and the inclusion and exclusion criteria have now been specified.
2. Optional: a simple summarising graphic would be a good addition to the paper.
Response: Thank you for the suggestion. Instead of a summarizing graphic, we have added a more informative figure that schematically lists, in the three main groups of factors, those that can affect the composition of breast milk.
Reviewer 2 Report
Comments and Suggestions for Authors
The authors have undertaken an interesting topic and made a commendable effort to evaluate factors that may affect human milk composition and energy value. Overall, the manuscript is well-written. The manuscript would benefit from clarifying the significance of each result, emphasising its relevance to child health and nutrient adequacy. In particular, several areas of concern require attention:
1) I recommend that authors review the terminology, for example, whether to use breast milk, human milk, or mother's milk in the manuscript. Please ensure consistency and use a single style throughout.
2) I would suggest specifying the abstract and Introduction parts, as they are very general;
3) There is no clear numbering of subsections in the Introduction part. Why is that?
4) It would be advisable to include the numbers of selected articles for the review.
5) Authors indicated that the search was limited to the period from May 2006 until May 2025. Older articles were not included because the methods used at that time to analyze the macronutrient and energy content of HM may not be as accurate as those used more recently, such as the Miris Human Milk Analyzer [8] launched in 2006, but the reason why the authors in Tables 5 and 6 reflect the polyunsaturated fat content is unclear. This cannot be determined with Miris Human Milk Analyser.
6) The discussion of results should be clearer and avoid replicating tables as text.
Author Response
1) I recommend that authors review the terminology, for example, whether to use breast milk, human milk, or mother's milk in the manuscript. Please ensure consistency and use a single style throughout.
Response: Thank you for your comment and suggestion. The terminology has been revised and is now consistent throughout the entire review. The revised manuscript only uses the term "breast milk."
2) I would suggest specifying the abstract and Introduction parts, as they are very general;
Response: Thank you for your comment and suggestion. The Abstract is limited to 200 words, therefore it is difficult to specify more results. Consequently, in the abstract, we chose to highlight the most consistent results. A new subsection, 1.1. "State of the Art," was added to the Introduction of the revised manuscript to address what is known about the topic and emphasize the relevance of the review.
3) There is no clear numbering of subsections in the Introduction part. Why is that?
Response: Thank you for your comment. In the former version and in revised version of the manuscript, the Introduction section include numbered subsections. In the revised Introduction, the five revised subsections are the following:
1.1. State of the Art
1.2. Collection and Analysis of Breast Milk
1.3. Methods of Feeding Preterm Infants Based on Breast Milk
1.4. Inaccuracy in Breast Milk Nutritional Content Estimate
1.5. Factors That May Affect the Macronutrient and Energy Content of Breast Milk
Don't hesitate to ask for further clarification if our answer is still unclear.
4) It would be advisable to include the numbers of selected articles for the review.
Response: Thank you for your comment and suggestion. The revised section 4 “Review Results” now specifies that 31 articles were included and critically analyzed.
5) Authors indicated that the search was limited to the period from May 2006 until May 2025. Older articles were not included because the methods used at that time to analyze the macronutrient and energy content of HM may not be as accurate as those used more recently, such as the Miris Human Milk Analyzer [8] launched in 2006, but the reason why the authors in Tables 5 and 6 reflect the polyunsaturated fat content is unclear. This cannot be determined with Miris Human Milk Analyser.
Response: Thank you for your comment and the opportunity to better explain this aspect. Since the Miris analyzer was launched in 2006 and has become the most widely used method, the time period for search from 2006 onwards was used to look for some uniformity in the studies using this method. Noteworthy, studies that used methods to determine the fatty acids and lactose content in breast milk were included regardless of this time criterion. This is now explained in the revised section 3 “Literature Search and Data Analysis”.
6) The discussion of results should be clearer and avoid replicating tables as text.
Response: Thank you for your comment. We consider that the information in the tables is summarized and essential. After addressing the results of the three groups of factors that can influence breast milk composition, a summary paragraph for interpreting the results in each of them has been included in the revised manuscript. Therefore, the main results are now interpreted in the body of the text and data organized in tables, avoiding replication of results.
Reviewer 3 Report
Comments and Suggestions for Authors
I’m concerned with the conclusion of the abstract “Higher protein and
energy content has been found in the milk from mothers who delivered
prematurely” as it perpetuates some small short-term differences as unconditionally important.
The abstract sentence is unclear and concerning: “Systematic reviews were prioritized, even though studies not included in these reviews and other types of literature reviews were also included” is not clear. How well do the authors recognize that study design is important but there are several other aspects of study quality (risks of bias) that are very important. It is likely that all breastmilk composition data is from observational studies (and meta-nalyses of the same), and therefore very prone to confounding by maternal and environment (how well were the women included or were better supported women with better determinants of health likely over-represented in the samples?) variables.
Author Response
I’m concerned with the conclusion of the abstract “Higher protein and
energy content has been found in the milk from mothers who delivered
prematurely” as it perpetuates some small short-term differences as unconditionally important.
Response: Thank you for your comment and the opportunity to better explain this aspect. In the revised Abstract it is now specified that this finding refers to the milk produced in the early stage of lactation.
The abstract sentence is unclear and concerning: “Systematic reviews were prioritized, even though studies not included in these reviews and other types of literature reviews were also included” is not clear. How well do the authors recognize that study design is important but there are several other aspects of study quality (risks of bias) that are very important. It is likely that all breastmilk composition data is from observational studies (and meta-nalyses of the same), and therefore very prone to confounding by maternal and environment (how well were the women included or were better supported women with better determinants of health likely over-represented in the samples?) variables.
Response: Thank you for your comment and the opportunity to explain the addressed topics. To be clearer, the sentence in the revised abstract has been rephrased as follows “Systematic reviews were prioritized, even though other types of literature reviews on the subject as well as studies not included in these reviews”. As this review is not a systematic review, a formal assessment of the quality of the studies (mostly observational) and the risk of bias was not carried out. This is acknowledged as a study limitation in the new section 5. “Strengths and Limitations” of the revised manuscript.
Reviewer 4 Report
Comments and Suggestions for Authors
The authors of this article collected, categorized, and described the results of publications on the factors affecting macronutrients and energy contents of human milk. No attempt was made to criticize, infer, nor synthesize the previous findings in this manuscript. So the manuscript may have a value as a list of publications on this topic: in this respect, indeed, this is a comprehensive collection of publications, but no novel, meaningful insight can be obtained from this manuscript.
I have no idea if this kind of publication collection is suitable for this journal.
The only point I’d like to address is:
Substantial introduction section is not given in this manuscript. This manuscript begins suddenly with methodological issues of milk sampling and analysis without no background information on the “review” topic. This is quite unusual for a scientific article.
Author Response
The authors of this article collected, categorized, and described the results of publications on the factors affecting macronutrients and energy contents of human milk. No attempt was made to criticize, infer, nor synthesize the previous findings in this manuscript. So the manuscript may have a value as a list of publications on this topic: in this respect, indeed, this is a comprehensive collection of publications, but no novel, meaningful insight can be obtained from this manuscript.
Response: Thank you for your comment and the opportunity to improve the manuscript. We consider that the most effective approach for building on previous knowledge was to include the most recent systematic reviews and other types of literature reviews on the subject, as well as studies not included in these reviews. To provide a novel insight from this review, after the discussion in the main three groups of factors, we have added a summary paragraph for the interpretation of each group of factors. To improve the critical analysis of the original studies, the respective levels of evidence were added. However, since this is not a systematic review, the following were not performed: data extraction strategy, assessment of study quality and risk of bias, and methods used for data synthesis and analysis.
I have no idea if this kind of publication collection is suitable for this journal.
The only point I’d like to address is:
Substantial introduction section is not given in this manuscript. This manuscript begins suddenly with methodological issues of milk sampling and analysis without no background information on the “review” topic. This is quite unusual for a scientific article.
Response: Thank you for your comment and suggestion and the opportunity to modify. A new subsection, 1. A new subsection, 1.1. "State of the Art," was added to the Introduction of the revised manuscript, to be more substantial and address what is known about the topic and emphasize the relevance of the review.
Round 2
Reviewer 3 Report
Comments and Suggestions for Authors
Thank you for your edits.
Your new abstract sentence: “Systematic reviews were prioritized, even though other types of literature reviews on the subject as well as studies not included in these reviews” seems to need something like “were included” at the end?
Please add “determinants of health” ((https://www.who.int/health-topics/social-determinants-of-health)) to your list of potentially important factors in your abstract and text. Here are some refs that observed that determinants of health are associated with breastfeeding success and whether mothers are able to express milk for their infants (PMID: 38807238 37395480 12873091 28244064 30170862), so likely better references could be found that they influence milk composition as well.
Author Response
Thank you for your edits.
Your new abstract sentence: “Systematic reviews were prioritized, even though other types of literature reviews on the subject as well as studies not included in these reviews” seems to need something like “were included” at the end?
Response: Thank you very much for the correction. The sentence has been completed by adding “were included”.
Please add “determinants of health” ((https://www.who.int/health-topics/social-determinants-of-health)) to your list of potentially important factors in your abstract and text. Here are some refs that observed that determinants of health are associated with breastfeeding success and whether mothers are able to express milk for their infants (PMID: 38807238 37395480 12873091 28244064 30170862), so likely better references could be found that they influence milk composition as well.
Response: Thank you very much for your valuable suggestion. We missed this important group of factors, and including them has substantially improved the review. Since social determinants of health impact breast milk composition via maternal health, we have included them in the "Maternal Factors" group. A sentence specific for the social determinants of health, as defined by the WHO, has been added to the section 3 “Literature Search and Data Analysis” of the revised manuscript. The manuscript and tables were updated to include the influence of social determinants of health on the composition of breast milk.
Reviewer 4 Report
Comments and Suggestions for Authors
Introduction section has been added to the original manuscript.
As I commented in the 1st round review, I don't know if this kind of "literature collection" article is suitable as a review for this journal: this is left for the Editor's judgement.
Author Response
Response: Thank you for your comment. The adequacy of including in Nutrients this "literature collection" in form of a critical review is certainly left to the Editor's judgment.
The authors chose this journal to publish their critical review because it publishes similar types of this review, such as the examples published during 2025:
- Poulios A, et al. The effects of antioxidant supplementation on soccer performance and recovery: a critical review of the available evidence. Nutrients. 2024 Nov 6;16(22):3803. doi: 10.3390/nu16223803.
- Cominelli G, et al. Neuro-nutritional approach to neuropathic pain management: a critical review. Nutrients. 2025 Apr 29;17(9):1502. doi: 10.3390/nu17091502.
- Bo S, et al. A critical review on the role of food and nutrition in the energy balance. 2020 Apr 22;12(4):1161. doi: 10.3390/nu12041161.